# The Therapeutic Potential of Naringenin: A Review of Clinical Trials

**DOI:** 10.3390/ph12010011

**Published:** 2019-01-10

**Authors:** Bahare Salehi, Patrick Valere Tsouh Fokou, Mehdi Sharifi-Rad, Paolo Zucca, Raffaele Pezzani, Natália Martins, Javad Sharifi-Rad

**Affiliations:** 1Student Research Committee, School of Medicine, Bam University of Medical Sciences, Bam 44340847, Iran; bahar.salehi007@gmail.com; 2Antimicrobial and Biocontrol Agents Unit, Department of Biochemistry, Faculty of Science, University of Yaounde 1, Ngoa Ekelle, Annex Fac. Sci., Yaounde 812, Cameroon; tsouh80@yahoo.fr; 3Department of Medical Parasitology, Zabol University of Medical Sciences, Zabol 61663-335, Iran; 4Department of Biomedical Sciences, University of Cagliari, 09042 Cagliari, Italy; 5OU Endocrinology, Dept. Medicine (DIMED), University of Padova, via Ospedale 105, 35128 Padova, Italy; 6AIROB, Associazione Italiana per la Ricerca Oncologica di Base, 35128 Padova, Italy; 7Faculty of Medicine, University of Porto, Alameda Professor Hernâni Monteiro, 4200-319 Porto, Portugal; 8Institute for Research and Innovation in Health (i3S), University of Porto, 4200-135 Porto, Portugal; 9Zabol Medicinal Plants Research Center, Zabol University of Medical Sciences, Zabol 61615585, Iran; 10Department of Chemistry, Richardson College for the Environmental Science Complex, The University of Winnipeg, 599 Portage Avenue, Winnipeg, MB R3B 2G3, Canada

**Keywords:** nutraceutics, phytochemicals, chemopreventive, cardiovascular diseases, flavonoids, citrus, flavanones

## Abstract

Naringenin is a flavonoid belonging to flavanones subclass. It is widely distributed in several *Citrus* fruits, bergamot, tomatoes and other fruits, being also found in its glycosides form (mainly naringin). Several biological activities have been ascribed to this phytochemical, among them antioxidant, antitumor, antiviral, antibacterial, anti-inflammatory, antiadipogenic and cardioprotective effects. Nonetheless, most of the data reported have been obtained from *in vitro* or *in vivo* studies. Although some clinical studies have also been performed, the main focus is on naringenin bioavailability and cardioprotective action. In addition, these studies were done in compromised patients (i.e., hypercholesterolemic and overweight), with a dosage ranging between 600 and 800 μM/day, whereas the effect on healthy volunteers is still debatable. In fact, naringenin ability to improve endothelial function has been well-established. Indeed, the currently available data are very promising, but further research on pharmacokinetic and pharmacodynamic aspects is encouraged to improve both available production and delivery methods and to achieve feasible naringenin-based clinical formulations.

## 1. Introduction

Naringenin is one of the most important naturally-occurring flavonoid, predominantly found in some edible fruits, like *Citrus* species and tomatoes [1,2,3], and figs belonging to smyrna-type *Ficus carica* [4]. Chemically named as 2,3-dihydro-5,7-dihydroxy-2-(4-hydroxyphenyl)-4H-1-benzopyran-4-one (Figure 1), naringenin shows a molecular weight of 272.26 (C_15_H_12_O_5_).

This widely distributed molecule is insoluble in water and soluble in organic solvents, like alcohol. Within the flavonoids class, naringenin is a flavanone that derives from naringin or narirutin (its glycone precursor) hydrolysis [5]. In fact, naringenin occupies a central position as primary C_15_ intermediate in the flavonoid biosynthesis pathway [6]. Naringenin biosynthesis has been investigated in *Medicago*, parsley and other plants. Overall, the metabolic pathway constitutes of a six step process, successively catalyzed by phenylalanine ammonia lyase (PAL), cinnamate 4-hydroxylase and its associated cytochrome P450 reductase, para-coumarate-CoA ligase, chalcone synthase (the key enzyme for the synthesis of naringenin) and chalcone isomerase [7]. Of note, naringenin is obtained by the condensation of para-coumaroyl-CoA with three units of malonyl-CoA. In addition, naringenin biosynthesis starter unit is para-coumaroyl-CoA, which in dicotyledonous plants derives from phenylalanine upon PAL deamination. The latter is thereafter hydroxylated at C_4_ by a cinnamate-4-hydroxylase and activated by a CoA-dependent ligase, through the universal phenylpropanoid pathway leading to flavonoids and stilbenes [8]. Moreover, monocotyledonous plants may also use tyrosine as substrate, directly producing *p*-coumaric acid without the need of cinnamate-4-hydroxylase enzyme activity [9,10].

To overcome the limited flavonoids production, in general, and naringenin, in particular, many attempts have been made to produce naringenin from metabolic engineering of specific pathways in microbial systems, such as *Escherichia coli* and *Saccharomyces cerevisiae* [7,10,11,12,13,14]. The highest naringenin titers obtained through biotransformation were reached using *E. coli* [7]. In addition, *S. cerevisiae* engineered to produce naringenin, solely from glucose using specific naringenin biosynthesis genes from *Arabidopsis thaliana*, led to flux optimization towards the naringenin pathway, providing a metabolic chassis for large amounts of naringenin production and biological functions exploration [7,11]. Growing evidence from both *in vitro* and *in vivo* animal studies have reinforced various naringenin pharmacological effects, including as hepatoprotective, anti-atherogenic, anti-inflammatory, anti-mutagenic, anticancer, antimicrobial agent, even suggesting its application in cardiovascular, gastrointestinal, neurological, metabolic, rheumatological, infectious and malignant diseases control and management [15,16,17,18]. Based on these aspects, the present review has a specific focus on clinical trials assessing naringenin consumptions’ health benefits, whereas the reported reviews are more related to other aspects (*in vitro* studies, over-production approaches, or specifically on oxidative stress only).

## 2. Preclinical Pharmacological Activities of Naringenin

Naringenin is endowed with broad biological effects on human health (Table 1), which includes a decrease in lipid peroxidation biomarkers and protein carbonylation, promotes carbohydrate metabolism, increases antioxidant defenses, scavenges reactive oxygen species, modulates immune system activity, and also exerts anti-atherogenic and anti-inflammatory effects [19,20]. It has also been reported to have a great ability to modulate signaling pathways related to fatty acids metabolism, which favors fatty acids oxidation, impairs lipid accumulation in liver and thereby prevents fatty liver [3], besides efficiently impairing plasma lipids and lipoproteins accumulation [21]. In addition, naringenin potentiates intracellular signaling responses to low insulin doses by sensitizing hepatocytes to insulin [19], besides being able to traverse the blood–brain barrier and to exert diverse neuronal effects, through its ability to interact with protein kinase C signaling pathways [19].

On the other hand, anti-cancer, anti-proliferative and anticarcinogenic effects have also been ascribed to this metabolite [22], mostly linked to its ability to repair DNA. In fact, cells exposition to 80 mM/L naringenin, during 24 h, led to 24% DNA hydroxyl damages reduction [20]. Moreover, antiviral effects have been reported. Naringenin shows a dose-dependent inhibitory effect against dengue virus [23], prevents intracellular replication of chikungunya virus [24], and inhibits assembly and long-term production of infectious hepatitis C virus particles in a dose-dependent manner [19]. Unfortunately, this bioflavonoid is poorly absorbed by oral ingestion, with only 15% of ingested naringenin absorbed in the human gastrointestinal tract [20], which has triggered several studies on its bioavailability.

## 3. Bioavailability of Naringenin

Naringenin bioavailability has been properly studied in previous works, suggesting an extensive pre-systemic gut flora metabolism, leading to a wide pattern of degradation products (i.e., phenolic acids) [79,80]. In a recent study, ultra-fast liquid chromatography-quadrupole-time-of-flight tandem mass spectrometry (UFLC-Q-TOF-MS/MS) was used to assess the urinary excretion of flavonoids in Chinese 23–30 years old volunteers, after 250 mL orange juice consumption (containing 31 μM naringenin). An overall 22% recovery was detected in 4 to 12 h, evidencing a phase II metabolism (especially sulfation and glucuronidation) of the aglycone after intestinal hydrolysis [81].

Bioavailability training effect was also investigated in male endurance athletes (clinicaltrails.gov NCT02627547) [82]. In this trial, 500 mL of orange juice (containing 76 μM naringenin) was ingested before and after 7 days of physical training cessation, and the urinary excretion of phenolic metabolites analyzed. As main findings, the authors stated that the bioavailability in endurance athletes was lower when compared with less trained individuals. However, short activity cessation slightly enhanced metabolites excretion [82]. In the same line, it was also shown that the urinary metabolites excretion does not differ after fresh oranges or pasteurized orange juice consumption, even if the latter contains about half of total flavanones amount [83].

Naringenin and hesperidin bioavailability were also investigated (trial NCT03032861) to deepen knowledge on orange juice prebiotic effect [84]. In this study, a marked increase in short-chain fatty acids and commensal bacteria were stated, with a concomitant decrease in ammonium levels, even in face of a decrease in total bacteria richness values.

## 4. Naringenin in Clinical Trials

Although there is a huge amount of data on *in vitro* biological effects of naringenin [85], only few clinical studies have been carried out [16], mainly because of the reduced data on pharmacokinetic aspects, metabolic fate and chemical instability of this compound [86]. Moreover, high isolation and purification costs further affect clinical trials feasibility.

Up to now, only 10 clinical studies were registered at clinicaltrials.gov database using “naringenin” or “naringin” (its glycoside) as keywords. Curiously, only one of these studies (NCT03582553, early phase I, still on recruiting) focused on naringenin administration isolated from *Citrus sinensis* extract (ranging from 150 to 900 mg). The main goal of this study was to check naringenin safety, tolerability and bioavailability, besides its effects on glucose metabolism. Data provided suggest how naringenin pharmacokinetics are still needing further investigation. Indeed, some of these studies investigated naringenin as a complex food supplement (i.e., whole orange juice), constituted by several polyphenols (including obviously naringenin), making difficult to assess the single phytochemicals contribution.

### 4.1. Role of Naringenin in Cardiovascular Diseases

The role of flavanones (including naringenin) on cardiovascular diseases has been well-studied [87], although most of the data have been collected in epidemiological and prospective studies. An inverse correlation has been stated between high flavanones consumption and cardiovascular risk [88,89,90,91], being a beneficial effect particularly related to naringenin consumption, given its great abundance in the tested samples [91]. In fact, most of the clinical studies have been carried out using naringin (a naringenin glycoside).

In a double-blind cross-over study, 12 patients with stage I hypertension received alternatively 500 mL/day of a fruit juice containing 593 μM naringin or a juice with lower content (143 μM naringin) for 5 weeks. Systolic blood pressure decreased in both groups, but no significant differences were found, while diastolic blood pressure was more effectively reduced in high-dose naringin group [92].

Dyslipidemic patients treated with a commercial bergamot-derived extract (containing about 95 μM naringin/capsule) evidenced plasmatic lipids reduction, while improved lipoprotein profile after 6 months [93]. The same glycoside was also able to decrease total plasma cholesterol levels and to enhance antioxidant defenses in hypercholesterolemic subjects [94]. Jung and colleagues prescribed 400 mg naringin/capsule/day, and after 8 weeks they also reported a decrease in LDL-cholesterol levels and an increase in some antioxidant enzymes activity (i.e., superoxide dismutase and catalase). A somewhat similar result was obtained in 237 hyperlipemic volunteers during a 30-day program, using a bergamot extract containing several flavonoids (including naringin). This plant extract preparation was able to decrease triglycerides, total and LDL cholesterol levels [95]. Quite surprisingly, in the study of Jung and colleagues, phytochemicals supplementation did not affect cholesterol levels in the healthy control group [94], but differently, in a randomized placebo-controlled trial including 194 moderately hypercholesterolemic patients [96], a daily dose of 1300 μM pure hesperidin or 862 μM pure naringin over 4 weeks, did not affect total or LDL cholesterol levels, this last result being in contrast to the work of Jung and collaborators. In fact, the authors suggested that the mean baseline LDL-cholesterol concentration in their study could not have contributed to the absence of LDL cholesterol effects and concluded that naringin (and hesperidin) did not have cholesterol-lowering effects when consumed as capsules. Certainly, this divergence should be further deepened; however, it appears that naringenin or naringin beneficial effects are closely related to patients with increased cardiovascular risk.

On the other hand, a clinical trial (NCT00539916) analyzed the effect of 600 mL/day orange juice consumption on 25 mild hypercholesterolemic male volunteers for 4 weeks [97]. The authors found some improvements in antioxidant profile and a tendency towards endothelial dysfunction decrease and slight increase in plasma apolipoprotein A-1 concentration. Similar results were also stated using whole orange juice in patients under hepatitis C antiviral therapy: Increase in antioxidant defenses, and decrease in inflammation and blood serum cholesterol levels [25]. The same research group achieved analogous effects on healthy volunteers, highlighting marked improvements in LDL-cholesterol and apolipoprotein B levels, and metabolic syndrome risk markers [98]. Another clinical trial (NCT03527277), although still in recruiting phase, is focused on whole orange juice effects in cardiovascular diseases- and type-2 diabetes-related metabolic markers, to be compared with sugar-sweetened beverages.

### 4.2. Role of Naringenin in Endothelial Function

Flow-mediated dilatation (FMD) of brachial artery at 0 to 7 h was used to assess the effect of 240 mL of orange flavanone beverages (about 15 mg naringenin) in a clinical trial involving 30–65 years healthy men [99]. Postprandial endothelial dysfunction was reduced, probably through a specific flavanone’s metabolites action on nitric oxide. Additionally, in a very interesting trial (NCT01272167), the long-term effect of 340 mL of grapefruit juice/day, containing about 480 μM naringenin glycoside, was investigated on endothelial function [67]. From the 48 healthy menopausal women recruited, arterial stiffness beneficial effects were found 6 months after treatment (carotid-femoral pulse wave velocity was significantly reduced).

### 4.3. Role of Naringenin in Weight Control

A commercial polyphenolic extract from several *Citrus* fruits (Sinetrol-XPur), containing about 20% of naringenin, was tested in 95 healthy overweight volunteers (BMI ranging from 26 to 29.9 kg/m^2^) [100]. The main overweight-related endpoints were improved after 12-weeks randomized protocol (including waist and hip circumference, abdominal fat, body weight). Moreover, inflammatory and oxidative stress markers were all decreased [100].

Stohs and coworkers also reported the naringin use as an adjuvant (600 mg) in weight management due to the well-known thermogenic effect of *Citrus aurantium* (bitter orange) extract (whose main active chemical compound is protoalkaloid *p*-synephrine) [101]. This double-blinded, randomized, placebo-controlled clinical trial (NCT01423019), involving 10 subjects per treatment group, showed that naringin is able to synergistically increase metabolic rate, without enhancing blood pressure and heart rates.

### 4.4. Role of Naringenin as Anti-HCV Activity

Naringenin has also been proposed as a novel therapeutic agent for hepatitis C virus (HCV) infection treatment. Indeed, this flavanone has been described to reduce HCV secretion in infected cells by 80%, at a concentration below to the toxic value in primary human hepatocytes and in mice [26]. Accordingly, in a phase I clinical trial already registered, 1 g naringenin supplementation (NCT01091077) was applied to examine its ability to hinder HCV infection and on very-low-density lipoproteins (vLDL) secretion lowering (usually acting as HCV carrier). Nevertheless, up to now, no published results are available. Based on the above described data, it emerges that, as no study has investigated naringenin chemopreventive potential on human cancer so far, this issue could be exploited in the near future.

However, might act by interfering with bioassays through several mechanisms and are termed Pan Assay INterference compoundS (PAINS) [102,103], which may affect the obtained bioassays results’ credibility and, thus, should be carefully analyzed [104].

## 5. Conclusions

Despite the huge amount of data on naringenin *in vitro* biological effects, few studies are available on its use as a therapeutic molecule. However, some specific effects were established under pure compounds supplementation, as well as in several studies using complex polyphenolic mixtures containing naringenin. The most promising activity seems to be related to cardiovascular disease protection, especially in already compromised patients. Nevertheless, these few data should be urgently expanded to better understand the naringenin mechanism of action on pathological or physiological conditions. However, a scarce number of clinical studies have been conducted so far, compromising its commercial exploitation. Further clinical studies are needed to better address naringenin safety, efficacy, delivery and bioavailability in humans.

## Figures and Tables

**Figure 1 pharmaceuticals-12-00011-f001:**
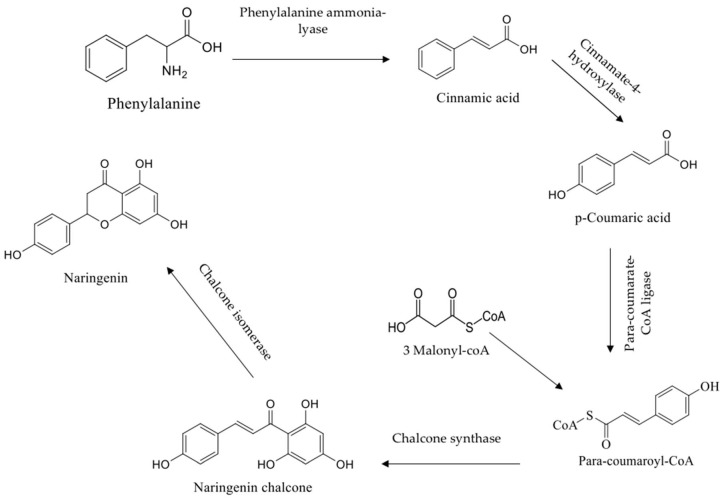
Naringenin biosynthesis.

**Table 1 pharmaceuticals-12-00011-t001:** Comparison of effective naringenin doses in various disease models.

Therapeutics	Diseases	Treatment	Targets and Effects	Route	Experimental Model	Ref.
Anti-Hepatitis C virus	Hepatitis C	2.7 mg/500 mL	Lipid profile and liver enzyme AST (decreased)	p.o.	Adult patients	[25]
		200 µM	Inhibition of apolipoprotein B secretion	-	*In vitro*, Huh7.5.1 human hepatoma cell	[26]
Antiaging	Aging-associated damage	4–40 μM	Reduction of senescence markers (X-gal, cell cycle regulator), oxidative stress (radical oxidative species, mitochondrial metabolic activity, mitochondrial calcium buffer capacity, estrogenic signaling functions)	-	*In vitro*, H9c2 embryonic rat cells	[27]
	Photoaging	1–4 MED (45 mJ/cm^2^)	Anti-photoaging effects by suppression of ERK2 activity and decrease of FRA1 stability, AP-1 transactivation and MMP-1 expression	-	*In vitro*, HaCaT keratinocyte cell line and the BJ human fibroblast cell	[28]
	Senescence process	50 mg/kg	Promotion of PI3K/Akt signaling, nuclear factor-erythroid 2-related factor 2, heme oxygenase 1, NAD(P)H-quinone oxidoreductase 1	p.o.	*In vivo*, mice	[29]
Anti-Alzheimer	Alzheimer	100 mg/kg	Mitigation of lipid peroxidation and apoptosis, attenuation of impairment of learning and memory	p.o.	*In vivo*, Wistar rats	[30]
Antiasthma	Asthma	9 mg/100 mL of the prepared fluid	Lowered subepithelial fibrosis, smooth muscle hypertrophy, and lung atelectasis	p.o.	*In vivo*, BALB/c mice	[31]
Anticancer	Breast cancer	250 µM	Inhibition of HER2-TK activity, anti-proliferative, pro-apoptotic and anti-cancerous activity	-	*In vitro*, SKBR3 and MDA-MB-231 breast tumor cells	[32]
	Liver cancer	100–200 μM	Block in G0/G1 and G2/M phase, accumulation of p53, apoptosis induction by nuclei damage, increased ratio of Bax/Bcl-2, release of cytochrome C, and sequential activation of caspase-3	p.o.	*In vitro*, human hepatocellular carcinoma HepG2 cells	[33,34]
	Postmenopausal breast cancer	High-fat (HF), high-fat diet with low naringenin (LN; 1% naringenin) or high-fat diet with high naringenin (HN; 3% naringenin)	Inhibition of cell growth, increases phosphorylation of AMP-activated protein kinase, down-regulation of CyclinD1 expression, and induction cell death. *In vivo*, delay of tumor growth (whereas no alteration of final tumor weight was observed)	p.o.	*In vitro*, E0771 mammary tumor cells. *In vivo*, ovariectomized C57BL/6 mice injected with E0771 cells	[17]
	Prostate cancer	5–50 μM	Inhibition of proliferation and migration, induction of apoptosis and ROS production. Loss of mitochondrial membrane potential and increased ratio of Bax/Bcl-2	-	*In vitro*, PC3 and LNCaP prostate cancer cells	[35]
	Melanoma	25–100 μM	Antiproliferative activity, increase of subG0/G1, S and G2/M phase cell proportion, decrease of cell proportion in G0/G1 phases	-	*In vitro*, B16F10 melanoma cells	[36]
	Gliomas-brain cancer	211 µM	Cytotoxicity	-	*In vitro*, human glioblastoma U-118 MG cells	[37]
	Breast cancer	200 mg/kg	Decreased secretion of TGF-β1 and accumulation of intracellular TGF-β1. Inhibition of TGF-β1 transport from the trans-Golgi network, and PKC activity	-	*In vivo*, Balb/c mice inoculated with breast carcinoma 4T1-Luc2 cells	[38]
Anti-Chikungunya virus	Chikungunya infection	6.818 µM	Inhibition of CHIKV intracellular replication	-	*In vitro*, CHIKV infected hamster kidney cells (BHK-21)	[24]
Anticonvulsant	Epilepsy	50–100 mg/kg	Inhibited production of TNFα and IL-1β, delaying the onset of seizures, and inhibiting activation of the mammalian target of rapamycin complex 1	p.o.	*In vivo*, male C57BL/6 mice injected with kainic acid	[39]
Anti-dengue virus	Dengue	250 μM	Prevention of infection	-	*In vitro*, dengue virus infected human-derived Huh7.5 hepatoma cell	[23]
Antidiabetic	Diabetic neuropathy	25–50 mg/kg	Attenuation of diabetic-induced changes in serum glucose, insulin and pro-inflammatory cytokines (TNF-alpha, IL-1beta, and IL-6). Attenuation of oxidative stress biomarkers. Decrease of insulin growth factor and nerve growth factor	p.o.	*In vivo*, streptozotocin-induced diabetic rats	[40]
	Diabetic retinopathy	50 mg/kg	Amelioration of oxidative stress, neurotrophic factors (brain derived neurotrophic factor (BDNF)), tropomyosin related kinase B (TrkB) and synaptophysin), and apoptosis regulatory proteins (Bcl-2, Bax, and caspase-3)	p.o.	*In vivo*, streptozotocin-induced diabetic rats	[41]
	Diabetes	0.05%	Improved glucose transporters (GLUTs 1, 3), and insulin receptor substrate 1 (IRS 1) levels	p.o.	*In vivo*, streptozotocin-induced diabetic rats	[42]
	Vascular endothelial dysfunction	50–100 mg/kg	Lowered levels of blood glucose, serum lipid, malonaldehyde, ICAM-1 and insulin resistance index, increased SOD activity and improved impaired glucose tolerance	p.o.	*In vivo*, streptozotocin-induced diabetic rats	[43]
	Diabetic renal impairment	5–10 mg/kg	Decrease in malondialdehyde levels, and affected superoxide dismutase, catalase and glutathione enzyme activities. Reduction in apoptosis activity, TGF-β1, and IL-1 expression	p.o.	*In vivo,* streptozotocin-induced diabetic rats	[44]
	Diabetes complications	50 mg/kg	Decreased lipid peroxidation level in liver and kidney tissue	p.o.	*In vivo,* alloxan-induced diabetic mice	[45]
Anti-Edwardsiellosis	Edwardsiellosis	200–400 µM	Down-regulation of *Edwardsiella tarda* infections	-	*In vitro*, Goldfish scale fibroblast (GAKS) cells	[46]
Anti-hyperlipidemic	Alcohol abuse, alcohol intolerance, alcohol dependence and other alcohol related disabilities	50 mg/kg	Decreased levels of plasma and tissue total cholesterol, triglycerides, free fatty acids, HMG CoA reductase and collagen content	p.o.	*In vivo*, male Wistar rats	[21]
Anti-inflammatory	Arthritic inflammation	5–20 mg/kg	Down-regulation of TNF-α, and NF-κB mRNA. Increased Nrf-2/HO-1s	p.o.	*In vivo*, Wistar rats	[47]
	Cognitive effect-memory impairment	25–100 mg/kg	Decreased expression of caspase-3, Bad, Bax, NF-κB, tumor necrosis factor-α, interleukin (IL)-6 and IL-1β	p.o.	*In vivo*, newborn Sprague-Dawley rats	[48]
	Endometriosis	5–100 µM	Antiproliferative and proapoptotic effect (Bax and Bak increased, activated MAPK and inactivated PI3K). Depolarization of mitochondrial membrane potential Activation of eIF2α and IRE1α, GADD153 and GRP78 proteins	-	*In vitro*, VK2/E6E7, vaginal mucosa derived epithelial endometriosis cells, and End1/E6E7, endocervix epithelial derived endometriotic cells	[49]
	Endotoxaemia	10 mg/kg	Suppression of TNF-α, IL-6, TLR4, inducible NO synthase (iNOS), cyclo-oxygenase-2 (COX2) and NADPH oxidase-2 (NOX2), NF-κB and mitogen-activated protein kinase (MAPK)	p.o.	*In vivo,* BALB/c mice In vitro, peritoneal macrophages obtained from the rats	[50]
	Hypertrophic scars (HS)	25–50 µM	Inhibition of hypertrophic scars. Downregulation of TNF-α, IL-1β, IL-6 and TGF-β1	p.o.	*In vivo*, female KM mice	[51]
	Liver diseases	50 mg/kg	Inhibition of oxidative stress, through TGF-β pathway and prevention of the trans-differentiation of hepatic stellate cells (HSC). Pro-apoptotic effect, inhibition of MAPK, TLR, VEGF, and TGF-β, Modulation of lipids and cholesterol synthesis.	p.o.	*In vivo*	[33]
	LPS-induced endotoxemia and Con A–induced hepatitis	100 μM 50 mg/kg 10 mg/kg	Post-translational inhibition of TNF-α and IL-6 (no interfering with TLR signaling cascade, cytokine mRNA stability, or protein translation)	- p.o. i.p.	*In vitro*, murine macrophage cell line RAW264.7 *In vivo*, female C57BL/6 mice *In vivo*, female BALB/c mice	[52]
	Lung injury	50–100 mg/kg	Down-regulation of nuclear factor-x03BA;B, inducible NO synthase, tumor necrosis factor-α, caspase-3; increased heat shock protein 70	p.o.	*In vivo*, rats	[53]
	Neuroinflammation-spinal cord injury	50–100 mg/kg	Repression of miR-223	p.o.	*In vivo*, female Wistar rats	[54]
	Osteoarthritis	40 mg/kg	Reduction in pain behavior and improvement in the tissue morphology. Inhibition of MMP-3 expression and NF-κB pathway	p.o.	*In vivo*, male Wistar rats	[55]
	Oxidative stress and lung damage	100 mg/kg	Reduction of oxidative stress, increase of antioxidant enzymes. Down-regulation of NF-κB, and COX-2	p.o.	*In vivo*, Wistar rats	[56]
	Pain	16.7–150 mg/kg	Analgesic effect, through activation of NO−cGMP−PKG−ATP-sensitive potassium channel pathway. Reduction of neutrophil recruitment, tissue oxidative stress, and cytokine production (IL-33, TNF-α, and IL-1β). Downregulation of mRNA expression of gp91phox, cyclooxygenase (COX)-2, and preproendothelin-1. Upregulation of nuclear factor (erythroid-derived 2)-like 2 (Nrf2) mRNA, and heme oxygenase (HO-1) mRNA expression, and NF-κB	p.o.	*In vivo*, male Swiss mice	[18,57]
	Protective effect on renal failure	50 mg/Kg	Improvement of renal markers. Decreased lipid profile and inhibition of pro-oxidant and inflammation markers	-	*In vivo*, rats	[58]
	Skin damage-burns	25–100 mg/kg	Inhibition of TNF-α, IL-1β, IL-6, NO, PGE2, caspase-3, LTB4 and NF-κB levels. Increased SOD, catalase, GPx and GST activities	p.o.	*In vivo*, male Wistar albino rats	[59]
Antimicrobial	Food-borne Staphylococcus aureus	0.92–3.68 mM)	Increased bacterial membrane permeability and changed cell morphology	-	*In vitro*, *Staphylococcus aureus* ATCC 6538	[60]
	*Escherichia coli*, *Staphylococcus aureus*, *Candida albicans*, *Alternaria alternata*, *Fusarium linii*, *Aspergillus niger*	OD in the range of 0–0.49 vs. 1.87 for controls	Antibacterial activity	-	*In vitro*, *Escherichia coli* ATCC10536, *Staphylococcus aureus* DSM799, *Candida albicans* DSM1386, *Alternaria alternata* CBS1526, *Fusarium linii* KB-F1, and *Aspergillus niger* DSM1957	[61]
Antioxidant	Skin injury	Pemphigus vulgaris (PV) serum treated HaCaT cell	Down-regulation of Dsg1, Dsg3, E-cadherin, ROS production, amelioration of the drop of mitochondrial membrane potential. Increase of the activity of SOD, GSH-Px and TAC. Decreased of NOD2, RIPK2 and NF-κB p-p65,	-	*In vitro*, human keratinocyte cell line HaCaT	[62]
Antiplatelet	Cardiovascular diseases	-	Antiplatelet activity targeting PAR-1, P2Y12 and COX-1 platelet activation pathways	-	*In silico*	[63]
Anti-stroke damage	Ischaemic stroke	20–80 µM	Inhibition of apoptosis and oxidative stress, and regulation of the localization of Nrf2 protein	p.o.	*In vivo/in vitro*, cortical neuron cells isolated from neonatal Sprague-Dawley rats	[64]
Cardioprotective	Cardiorenal syndrome	50 mM; 25–50 mg/kg	Attenuation of cardiac remodeling and cardiac dysfunction, decrease of left ventricle weight (LVW), increase of body weight (BW), decrease of LVW/BW, blood urea, type-B natriuretic peptide, aldosterone, angiotensin (Ang) II, C-reactive protein	p.o.	*In vivo*, male Sprague Dawley rats *In vitro,* cardiac fibroblasts	[65]
	Hypoxia/reoxygenation (H/R) injury	80 µM	Overexpression of Bcl-2, glucose-regulated protein 78, cleaved activating transcription factor 6 (ATF6) and phosphorylation levels of phospho-extracellular regulated protein kinases (PERK). Decrease of caspase-3, and Bax	-	*In vitro*, rat cardiomyocyte H9c2 cells	[66]
	Arterial stiffness in postmenopausal	210 mg/day	Decreased carotid-femoral pulse wave velocity	p.o.	Patients, healthy postmenopausal women	[67]
	Atherosclerosis and coronary heart diseases	200 µM	Upregulation of SREBP-1a promoter activity	-	*In vitro*, human hepatoma HepG2 cells	[68]
Chronic kidney disease	Renal fibrosis/ obstructive nephropathy	50 mg/kg	Inhibition of Smad3 phosphorylation and transcription	p.o.	*In vivo*, C57BL6 male mice	[69]
Expectorant	Sputum symptoms	100 µM	Increase of CFTR expression, stimulation of chloride anion secretion	apical	*In vivo*, Sprague-Dawley rats	[70]
Eye-protective	Corneal neovascularization	0.08–80 µg; 8 µL of 0.01–10 g/L solution	Inhibition of alkali burn-induced neutrophil (myeloperoxidase activity and recruitment of Lysm-GFP+ cells) and macrophage (N-acetyl-β-D glycosaminidase activity) recruitment. Inhibition of IL-1β., IL-6 production, Vegf, Pdgf, and Mmp14 mRNA expression	Eye drop	*In vivo*, male Swiss mice	[71]
Fertility	Infertility	40–80 mg/kg	Attenuation of DNA fragmentation and sperm count during antiretroviral therapy	p.o.	*In vivo*, male Sprague-Dawley rats	[72]
Immunomodulatory	Immunodepression	5.4–21.6 μg/mL	Increase of B cell proliferation, and NK activity	-	*In vitro*, spleen mice lymphocytes and peritoneal macrophages obtained from pathogen-free male BALB/c mice	[73]
Laxative	Constipation	75–300 mg/kg	Amelioration of constipation, increased c-Kit, SCF, and aquaporin 3	p.o.	*In vivo*, ICR mice	[15]
Hepatoprotective	Alcoholic liver disease/steatosis	2.5–10 mg/kg	Reduction of alcohol-related gene expression (cyp2y3, cyp3a65, hmgcra, hmgcrb, fasn, fabp10α, fads2 and echs1)	-	In vivo, zebrafish larvae	[74]
	Hepatitis B virus protein X (HBx)-induced hepatic steatosis	30 mg/kg	Down-regulation of SREBP1c, LXRα, and PPARγ genes	p.o.	*In vivo*, HBx-transgenic C57BL/6 mice *In vitro*, HBx-transfected human hepatoma HepG2 cells	[75]
Pregnancy	Migration mechanism(s) of peri-implantation conceptuses	20 µM	Stimulation of pTr cells migration, through PI3K/AKT and ERK1/2 MAPK signaling pathways	-	*In vitro*, porcine trophectoderm (pTr) cells	[76]
Radioprotective	Radiation-induced DNA, chromosomal and membrane damage.	50 mg/kg	Inhibition of NF-kB pathway, apoptotic proteins: p53, Bax, Bcl-2	p.o.	*In vivo*, Swiss albino mice	[77]
Weight loss	Obesity: Muscle loss and metabolic syndrome in postmenopausal women.	3% naringenin diet	Down-regulation of genes involved in de novo lipogenesis, lipolysis and triglyceride synthesis/storage	p.o.	*In vivo*, C57BL/6J mice	[78]

MED, minimal erythema dose.

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
