# Peer review of "The Therapeutic Potential of Naringenin: A Review of Clinical Trials"

_pharmaceuticals, 2019, doi:10.3390/ph12010011_

Round 1

Reviewer 1 Report

This paper concerns the activity of naringenin, a flavonoid compound which presents interesting bioactivities in health and diseases. The most interesting point of this work concerns the description of numerous clinical trials which have already been published or are under progress.This view is well written and esay to read. It will be useful for the researchers involved in this area of investigation.

I have some remarks of real concern :

Figure 1:

Replace 4-coumaric acid CoA ligase by para-coumarate CoA ligase

Replace 4-coumaroyl-CoA by para-coumaroyl-CoA

Please add: Chalcone synthase, the key enzyme for the synthesis of naringenin

Naringenin is obtained by the condensation of para-coumaroyl-CoA with three units of malonyl-CoA. This should be indicated since the malonyl-Coa units are missing.

Lines 53-54: I don't understand this sentence. Do you mean that naringenin inhibits by a feed-back effect the p-coumaroyl-CoA ligase ?

Lines 54 and 57: write para-coumarate

Line 58: write para-coumaroyl-CoA

Line 61: complete the sentence "...by a CoA-dependent ligase, through the universal phenylpropanoid pathway leading to flavonoids and stilbenes (Jeandet et al., 2018)".

and add the following reference: Jeandet P., Sobarzo-Sánchez E., Clément C., Nabavi S.F., Habtemariam S., Nabavi S.M. and Cordelier S. Engineering stilbene metabolic pathways in microbial cells. Biotechnology Advances, 2018, 36, 2264-2283

Line 65: to references 7,9,10, please add the following references:

Pandley, R.P.; Parajuli, P.; Koffas, M.A.G.; Sohng, J.K., 2016. Microbial production of natural and nonnatural flavonoids: Pathway engineering, directed evolution and systems/synthetic biology. Biotech. Adv. 34, 634-662.

Trantas, E.; Koffas, M.A.G.; Xu, P.; Ververidis, F., 2015. When plants produce not enough or at all: metabolic engineering of flavonoids in microbial hosts. Front. Plant Sci. 6:7.

Nabavi S.M., Shirooie S., Samec D., Tomczyk M., Milella L., Russo D., Habtemariam S., Suntar I., Rastrelli L., Daglia M., Xia J., Giampieri F., Battino M., yousefi B., Sobarzo-Sánchez E., Jeandet P., Xu S., and Nabavi S.F. Flavonoid biosynthetic pathways in plants: versatile targets for metabolic engineering. Biotechnology Advances, 2018, in press, https://doi.org/10.1016/j.biotechadv.2018.11.005

Line 124

Line 70: have reinforced

Line 124: "...instability of this compound"

Line 130: Data provided suggest

Author Response

Comments and Suggestions for Authors

This paper concerns the activity of naringenin, a flavonoid compound which presents interesting bioactivities in health and diseases. The most interesting point of this work concerns the description of numerous clinical trials which have already been published or are under progress.This view is well written and esay to read. It will be useful for the researchers involved in this area of investigation.

I have some remarks of real concern :

Figure 1: 

Replace 4-coumaric acid CoA ligase by para-coumarate CoA ligase

Replace 4-coumaroyl-CoA by para-coumaroyl-CoA

Please add: Chalcone synthase, the key enzyme for the synthesis of naringenin

Naringenin is obtained by the condensation of para-coumaroyl-CoA with three units of malonyl-CoA. This should be indicated since the malonyl-Coa units are missing.

Answer: We thank the reviewer for the amendments. These corrections have been now included in the manuscript.

Lines 53-54: I don't understand this sentence. Do you mean that naringenin inhibits by a feed-back effect the p-coumaroyl-CoA ligase? 

Answer: The part “at same time that inhibits the key enzyme activity of the phenylpropanoid pathway, 4-coumarate CoA ligase” has been deleted because in its actual form could lead to misunderstanding.

Lines 54 and 57: write para-coumarate

Line 58: write para-coumaroyl-CoA

Line 61: complete the sentence "...by a CoA-dependent ligase, through the universal phenylpropanoid pathway leading to flavonoids and stilbenes (Jeandet et al., 2018)". 

and add the following reference: Jeandet P., Sobarzo-Sánchez E., Clément C., Nabavi S.F., Habtemariam S., Nabavi S.M. and Cordelier S. Engineering stilbene metabolic pathways in microbial cells. Biotechnology Advances, 2018, 36, 2264-2283

Line 65: to references 7,9,10, please add the following references:

Pandley, R.P.; Parajuli, P.; Koffas, M.A.G.; Sohng, J.K., 2016. Microbial production of natural and nonnatural flavonoids: Pathway engineering, directed evolution and systems/synthetic biology. Biotech. Adv. 34, 634-662.

Trantas, E.; Koffas, M.A.G.; Xu, P.; Ververidis, F., 2015. When plants produce not enough or at all: metabolic engineering of flavonoids in microbial hosts. Front. Plant Sci. 6:7.

Nabavi S.M., Shirooie S., Samec D., Tomczyk M., Milella L., Russo D., Habtemariam S., Suntar I., Rastrelli L., Daglia M., Xia J., Giampieri F., Battino M., yousefi B., Sobarzo-Sánchez E., Jeandet P., Xu S., and Nabavi S.F. Flavonoid biosynthetic pathways in plants: versatile targets for metabolic engineering. Biotechnology Advances, 2018, in press,https://doi.org/10.1016/j.biotechadv.2018.11.005

Answer: The corrections suggested by the reviewer have been made and the new references added to the manuscript.

Line 124

Line 70: have reinforced

Line 124: "...instability of this compound"

Line 130: Data provided suggest

Answer: The corrections suggested by the reviewer have been made in the text.

Reviewer 2 Report

-  Fig 1 should be cited in the text.

-  Line 79: remove reference 1 or explain its need since it seems that Table 1 was based in the studies mentioned in reference 1.

-  Table 1 should be improved since the studies are not discussed in detail in the text. I suggest that authors replace “effective dose” by “treatment” where they should also detail the period of treatment. Besides, “effects” should be detailed, instead of “Targets” and “Remarks” should also refer the specific experimental model.

-  Data related to clinical trials should be summarized in a Table.

-  Authors should try to distinguish their review from those previously reported (eg refences 83 and 12).

Author Response

Comments and Suggestions for Authors

-  Fig 1 should be cited in the text.

Answer: Fig. 1 is cited at page 1 (Introduction) line 44.

-  Line 79: remove reference 1 or explain its need since it seems that Table 1 was based in the studies mentioned in reference 1.

Answer: We followed the reviewer’s suggestion and deleted the reference 1 from the text.

-  Table 1 should be improved since the studies are not discussed in detail in the text. I suggest that authors replace “effective dose” by “treatment” where they should also detail the period of treatment. Besides, “effects” should be detailed, instead of “Targets” and “Remarks” should also refer the specific experimental model.

Answer: We replaced the “effective dose” by “treatment” and detailed as required by the reviewer.

-  Data related to clinical trials should be summarized in a Table.

Answer: We decided not to summarize the clinical trials data because we wanted to avoid the use of 2 tables in the manuscript.

-  Authors should try to distinguish their review from those previously reported (eg refences 83 and 12).

Answer: The criticism of the Reviewer is quite understandable, since naringenin’s health effect is a topic of huge interest by the scientific community, leading to a continuous flow of papers in the last few years. For this reason, however, we think that the need to give order to this enormous amount of data is high. Besides, our MS has a specific focus on clinical trials, whereas the reported reviews are more related to other aspects (in vitro studies, over-production approaches, or specifically on oxidative stress only).

Round 2

Reviewer 1 Report

My comments have satisfactorily been addressed.

I only have three remarks of real concern:

Why is there a square on the malonyl-CoA structure (Figure 1) ? Please correct. Check also why there are spaces within the names of some enzymes.

Line 253: reference 8 is not complete (the page number is lacking). Please check and correct as follows;

Jeandet P., Sobarzo-Sánchez E., Clément C., Nabavi S.F., Habtemariam S., Nabavi S.M. and Cordelier S. Engineering stilbene metabolic pathways in microbial cells. Biotechnology Advances, 2018, 36, 2264

Line 268: reference 14 is not complete (authors are lacking). Please correct as follows:

Nabavi S.M., Shirooie S., Samec D., Tomczyk M., Milella L., Russo D., Habtemariam S., Suntar I., Rastrelli L., Daglia M., Xia J., Giampieri F., Battino M., yousefi B., Sobarzo-Sánchez E., Jeandet P., Xu S., and Nabavi S.F. Flavonoid biosynthetic pathways in plants: versatile targets for metabolic engineering. Biotechnology Advances, 2019, in press,https://doi.org/10.1016/j.biotechadv.2018.11.005.

Author Response

Why is there a square on the malonyl-CoA structure (Figure 1) ? Please correct. Check also why there are spaces within the names of some enzymes.

Answer: The required change in Figure 1 was properly done.

Line 253: reference 8 is not complete (the page number is lacking). Please check and correct as follows;

Jeandet P., Sobarzo-Sánchez E., Clément C., Nabavi S.F., Habtemariam S., Nabavi S.M. and Cordelier S. Engineering stilbene metabolic pathways in microbial cells. Biotechnology Advances, 2018, 36, 2264

Answer: The required change was done.

Line 268: reference 14 is not complete (authors are lacking). Please correct as follows:

Nabavi S.M., Shirooie S., Samec D., Tomczyk M., Milella L., Russo D., Habtemariam S., Suntar I., Rastrelli L., Daglia M., Xia J., Giampieri F., Battino M., yousefi B., Sobarzo-Sánchez E., Jeandet P., Xu S., and Nabavi S.F. Flavonoid biosynthetic pathways in plants: versatile targets for metabolic engineering. Biotechnology Advances, 2019, in press,https://doi.org/10.1016/j.biotechadv.2018.11.005

Answer: The required change was done.

Reviewer 2 Report

Although I understand that naringenin’s health effect is a topic of huge interest and leads to a continuous flow of papers, I believe that repetitive reviews are not high relevant for readers. In my view, authors should emphasize their approach, which they state to be different from the previous ones “our MS has a specific focus on clinical trials, whereas the reported reviews are more related to other aspects (in vitro studies, over-production approaches, or specifically on oxidative stress only)”.

Author Response

Although I understand that naringenin’s health effect is a topic of huge interest and leads to a continuous flow of papers, I believe that repetitive reviews are not high relevant for readers. In my view, authors should emphasize their approach, which they state to be different from the previous ones “our MS has a specific focus on clinical trials, whereas the reported reviews are more related to other aspects (in vitro studies, over-production approaches, or specifically on oxidative stress only)”.

Answer: Thank you for your comment. A specific sentence was included in introduction section (l. 77-80).